# Seasonal and comparative evidence of adaptive gene expression in mammalian brain size plasticity

**William R Thomas**[1]*, **Troy Richter**[2], **Erin T O'Neil**[2], **Cecilia Baldoni**[3,4],
**Angelique Corthals**[5], **Dominik von Elverfeldt**[6], **John D Nieland**[7], **Dina Dechmann**[3,4],
**Richard Hunter**[2], **Liliana M Davalos**[1,8]

[1]Department of Ecology and Evolution, Stony Brook University, New York, United States; [2]Department of Psychology, Developmental and Brain Sciences Program, University of Massachusetts Boston, Boston, United States; [3]Max Planck Institute of Animal Behavior, Radolfzell, Germany; [4]University of Konstanz, Radolfzell, Germany; [5]John Jay College of Criminal Justice, New York, United States; [6]Division of Medical Physics, Department of Dignostic and Interventional Radiology, University Medical Center Freiburg, Faculty of Medicine, University Freiburg, Freiburg, Germany; [7]Health Science and Technology, Aalborg University, Aalborg, Denmark; [8]Consortium for Inter-Disciplinary Environmental Research, Stony Brook University, New York, United States

*For correspondence:
William.thomas@stonybrook.edu

## eLife Assessment

This study presents **valuable** findings related to seasonal brain size plasticity in the Eurasian common shrew (Sorex araneus), which is an excellent model system for these studies. The evidence supporting the authors' claims is **convincing**. The work will be of interest to biologists working on neuroscience, plasticity, and evolution.

**Abstract** Contrasting almost all other mammalian wintering strategies, Eurasian common shrews, *Sorex araneus*, endure winter by shrinking their brain, skull, and most organs, only to then regrow to breeding size the following spring. How such tiny mammals achieve this unique brain size plasticity while maintaining activity through the winter remains unknown. To discover potential adaptations underlying this trait, we analyzed seasonal differential gene expression in the shrew hypothalamus, a brain region that both regulates metabolic homeostasis and drastically changes size, and compared hypothalamus gene expression across species. We discovered seasonal variation in suites of genes involved in energy homeostasis and apoptosis, shrew-specific upregulation of genes involved in the development of the hypothalamic blood-brain barrier and calcium signaling, as well as overlapping seasonal and comparative gene expression divergence in genes implicated in the development and progression of human neurological and metabolic disorders, including *CCDC22*. With high metabolic rates and facing harsh winter conditions, *S. araneus* have evolved both adaptive and plastic mechanisms to sense and regulate their energy budget. Many of these changes mirrored those identified in human neurological and metabolic disease, highlighting the interactions between metabolic homeostasis, brain size plasticity, and longevity.

## Introduction

Typical mammalian brain development consists of unidirectional postnatal growth until reaching an adulthood maximum (*Arain et al., 2013*; *Gould, 1966*), but the Eurasian common shrew, *Sorex araneus*, seasonally changes the size of its body, most organs, skull, and—especially—its brain (*Lázaro et al., 2019*; *Pucek, 1970*; *Pucek, 1965b*; *Pucek, 1965a*). In the most acute case of Dehnel's phenomenon, or seasonal size plasticity, the common shrew first grows to an initial summer maximum as a juvenile, then shrinks in autumn losing up to 18% body mass, ~20% skull volume, and 26% brain mass, reaching its winter size minimum (*Lázaro et al., 2019*; *Pucek, 1970*; *Pucek, 1965b*; *Pucek, 1965a*). Then, in the spring prior to mating, these shrews partially or fully regrow these organs, reaching a second maximum. Dehnel's phenomenon is hypothesized to be an adaptation to decrease energy demand without slowing metabolism through plasticity, improving shrew fitness during the low temperatures and limited food supply of winter (*Lázaro et al., 2019*; *Pucek, 1970*; *Pucek, 1965a*; *Keicher et al., 2017*; *Churchfield et al., 2012*; *Taylor et al., 2013*; *Hyvärinen, 1984*; *Schaeffer et al., 2020*; *Thomas et al., 2023*). Corroborating this adaptive plasticity hypothesis, Dehnel's phenomenon anticipates winter conditions and is both geographically variable (*Pucek, 1970*) and modulated by environmental temperature (*Lázaro et al., 2019*). Although we have begun to reveal the molecular mechanisms of Dehnel's phenomenon (*Thomas et al., 2023*), the comparative context of proposed adaptations and commonality of processes across brain regions remain unknown. Elucidating both the genetic basis and evolutionary mechanisms underlying Dehnel's phenomenon will illuminate the molecular adaptations involved in this unusual case of mammalian brain plasticity.

As a hypothesized wintering strategy, energy metabolism is key to the evolution of Dehnel's phenomenon in *S. araneus* (*Schaeffer et al., 2020*). Tiny mammals have limited options to survive winter conditions (*Auteri and Knowles, 2020*), and Dehnel's phenomenon is exceedingly rare as a wintering strategy. While many mammals seasonally migrate away from low-temperature, low-productivity environments, some endure these conditions by reducing their energy requirements through hibernation, a phenotype that has independently evolved many times (*Geiser, 2008*; *Ferris and Gregg, 2019*; *Turbill et al., 2011*). The evolution and regulation of hibernation relies upon a suite of metabolic genes (*Faherty et al., 2016*; *Villanueva-Cañas et al., 2014*), including those associated with thermogenesis, circadian rhythms, and feeding behavior. Compared to most species, *S. araneus* has astronomically large energy demands, with one the highest basal metabolic rates per unit of body mass identified in any mammal (*Genoud et al., 2018*; *Taylor, 1998*). To meet these energy requirements, *S. araneus* forage and feed constantly. Together with brief lifespans (~1 year) that nonetheless require surviving a single winter (*Searle et al., 2019*; *Healy et al., 2014*), these extreme metabolic demands constrain survival options for *S. araneus*. Thus, by decreasing resources devoted to energetically expensive tissue such as the brain (*Mergenthaler et al., 2013*), Dehnel's phenomenon bypasses these constraints and expands the niche of a typical small mammal during winter while allowing this fast-living, predatory shrew to remain active year round.

We hypothesize Dehnel's phenomenon, which involves metabolic shifts and interorgan communication (*Thomas et al., 2023*), has evolved through the combination of both physiological plasticity and genetic adaptation. This idea is twofold: phenotypic plasticity is not only adaptive in itself as the evolved spectrum of a trait, but can also allow populations to persist in varying environments, leading to ancillary evolutionary adaptation when given enough time and strength of selection (*Via and Lande, 1985*). Thus, while temporal analysis of shrew gene expression can elucidate putatively adaptive plastic regulation, it will miss adaptive canalization of gene expression that has contributed to the evolution of Dehnel's phenomenon. Therefore, plastic regulatory mechanisms and proposed adaptations can be further tested via cross-species comparisons of gene divergence. These cross-species comparisons treat gene expression as a trait and have been used to characterize diverse vertebrate adaptations including thermal tolerance and vision loss in fish (*Brauer et al., 2017*; *Bernal et al., 2020*; *Stern and Crandall, 2018*), the effects of whole genome duplication on gene expression (*Gillard et al., 2021*; *Conant, 2020*; *Braasch et al., 2016*; *Lien et al., 2016*; *Sandve et al., 2018*), sex-specific alternative splicing in birds (*Rogers et al., 2021*), and patterns of gene expression evolution in mammalian organs (*Rohlfs et al., 2015*; *Chen et al., 2019*; *Yapar et al., 2021*; *Sjöstedt et al., 2020*). By incorporating these comparative approaches with seasonally varying gene expression, we can further elucidate the role of plasticity and adaption in both the regulation and evolution of Dehnel's phenomenon.

The hypothalamus is an intriguing candidate brain region for comparative analysis, as it is both the brain region with the greatest seasonal size change in shrews and the center of bodily homeostatic maintenance across mammals. In shrews, the hypothalamus undergoes a 31.6% volume reduction in autumn through winter, followed by a 47.8% increase in spring (*Lázaro et al., 2018*). Across all mammals, the hypothalamus plays a pivotal role in the maintenance of energy budgets, with functions influencing: (1) energy intake and feeding behavior, (2) energy expenditure and metabolic rate, and (3) energy deficits and storage. While such functions are important for all mammals, these are critical while wintering to maintain stable internal body temperatures as external temperatures decrease. For example, the hypothalamus can activate the sympathetic nervous system as temperatures decrease, stimulating winter thermogenesis. Thus, shrews may deploy typical mammalian hypothalamic plasticity concurrent with seasonal size plasticity. Alternatively, and considering how rare Dehnel's phenomenon is, shrews may have evolved divergent approaches, akin to naked mole rat traits, a species that lives under consistent hypoxic conditions resulting in adaptations associated with decreased oxygen and blood flow across the brain (*Pamenter, 2022*). Comparative approaches, then, may identify novel adaptations that contribute to the evolution of Dehnel's phenomenon in the shrew hypothalamus.

As the brain region able to quickly respond to change, the hypothalamus has remarkable plastic capabilities associated with metabolism (*Dietrich and Horvath, 2013*), especially as environmental conditions vary between seasons and can be unpredictable. While the central nervous system (CNS) acts somewhat independently of bodily metabolism through the protective effects of the blood-brain barrier (BBB), the CNS is still sensitive to bodily metabolic change. We hypothesize hypothalamic plasticity is central to the evolution and regulation of size plasticity through both environmental sensing mechanisms and signaling responses to these stimuli. Specifically, we hypothesize adaptations in the shrew: (1) hypothalamic BBB associated with dynamic metabolic fluctuations, (2) sensing of metabolic state and seasonal size through hormonal signals, mediated by BBB crossing of insulin, ghrelin, and leptin, and (3) responses to metabolic fluctuations involving ion-dependent signaling. With limited energy inertia, shrews must continuously sense their peripheral metabolism, and a combination of receptors and selective BBB permeability in the hypothalamus allows certain molecules to relay information from the peripherals to the brain. Among the best characterized of these are the peripheral hormones insulin, ghrelin, and leptin, the latter known to excite anorexigenic and inhibit orexigenic neurons in the arcuate nucleus of the hypothalamus. Finally, according to Ramon y Cajal's neuronal doctrine (*Ramón y Cajal, 1891*), differences in brain functional responses may stem from altered synaptic firing. Thus, inter- and intracellular ion concentrations play a large role in the communication between neuronal networks by propagating signals in response to perceived environmental stimuli.

To test these hypotheses, we analyzed both the seasonal and phylogenetic variation in shrew hypothalamic gene expression to detect signals of adaptive plasticity. First, we aimed to identify differential gene expression of genes across five seasons of Dehnel's phenomenon that might promote regulatory responses to seasonal variation and can be functionally validated with cell line perturbations. Second, this analysis was paired with a comparative transcriptomics approach using hypothalamic gene expression data from 15 additional mammal species. These analyses infer putative adaptation by testing for branch-specific gene expression shifts using Ornstein-Uhlenbeck models. The objective of these evolutionary analyses was to quantify lineage-specific hypothalamic gene expression changes in *S. araneus*, with evolutionary gene expression divergence consistent with selection. Finally, by comparing individual genes and associated pathways from these two analyses, we can determine potential adaptive plasticity, indicating mechanisms that both potentially regulate size plasticity and were selected for higher gene expression in the evolution of Dehnel's phenomenon. Our results implicate several key processes in Dehnel's phenomenon, including seasonal plasticity in feeding behavior, enhanced modulation of both hypothalamic BBB and downstream signaling, and plastic apoptosis responses.

## Results

### Temporal gene expression

We identified several known neural signaling pathways correlated with patterns of seasonal hypothalamus gene expression that may be associated with changes during Dehnel's phenomenon. We began by hierarchically clustering gene expression to identify patterns through time. Most genes

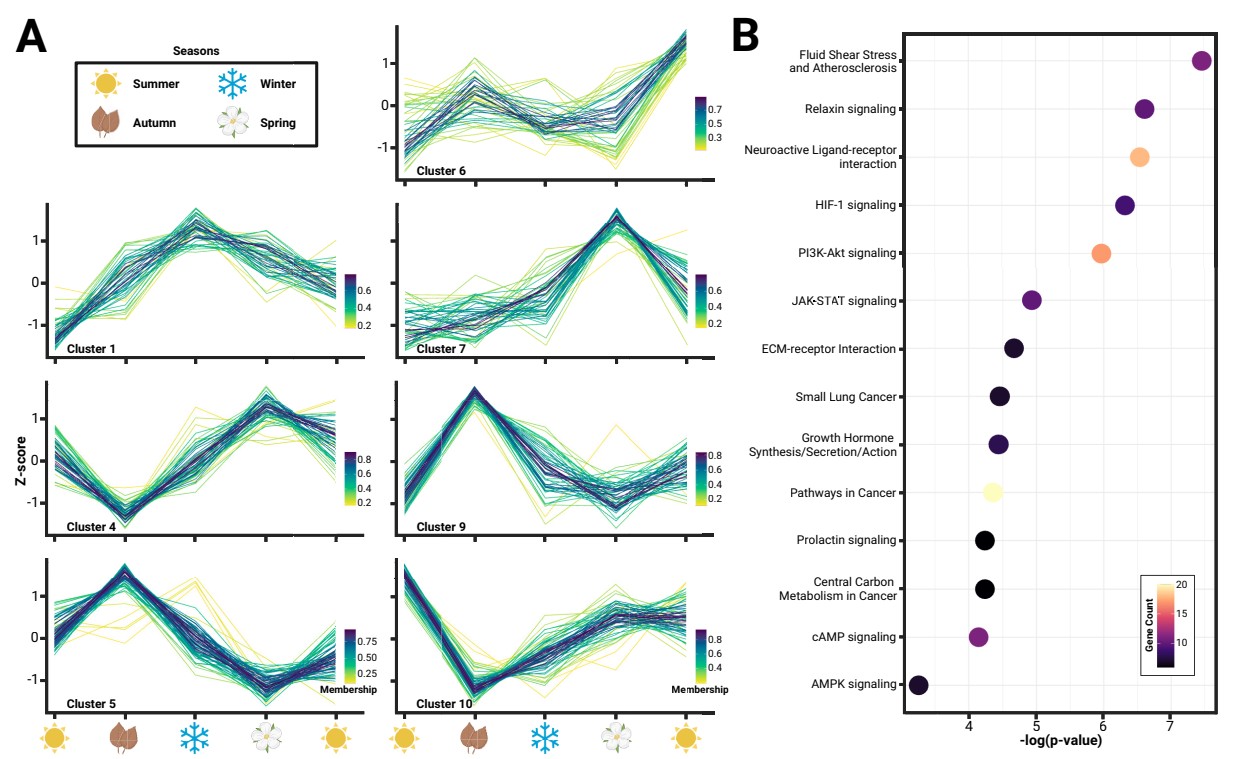

**Figure 1.** Seasonal changes in hypothalamic gene expression enriches pathways related to homeostasis. (**A**) Hierarchical clustering of gene expression identified 12 distinct clusters, of which 7 clusters (*Arain et al., 2013*; *Pucek, 1970*; *Pucek, 1965b*; *Pucek, 1965a*; *Keicher et al., 2017*; *Taylor et al., 2013*; *Hyvärinen, 1984*) comprising of 394 genes, showed variation consistent with seasonality or Dehnel's phenomenon. (**B**) Functional characterization of these genes using Kyoto Encyclopedia of Genes and Genomes (KEGG) Gene Ontology (GO) pathways found an enrichment of 14 pathways (p<0.05), many of which have been implicated in hypothalamic control of homeostatic maintenance, including relaxin signaling, neuroactive ligand-receptor interaction, HIF-1 signaling, and phosphoinositide-3 kinase (PI3K)-Akt signaling.

The online version of this article includes the following figure supplement(s) for figure 1:

**Figure supplement 1.** Hierarchical clustering of 786 gene expression profiles formed 12 distinct clusters in the shrew hypothalamus.

did not exhibit much variation through time, indicating constitutive gene expression in the hypothalamus, as only 786 of 19,296 genes passed our filters (>0.5-fold change between any two seasons and 2 samples>10 normalized reads). We identified 12 distinct clusters of gene expression patterns by bootstrapping (n=20) a gap statistic within cluster distances. Of these 12 clusters, five clusters consisting of 392 genes resembled a large divergence between summer juveniles and the remaining individuals without a second local maximum during size change (*Figure 1—figure supplement 1*; Clusters 2, 3, 8, 11, 12). These genes likely represent a large developmental shift between recently postnatal shrews and the remainder of individuals compared to shifts associated with seasonality or Dehnel's phenomenon. We then functionally characterized the remaining 394 genes (*Figure 1A*), which exhibited seasonal shifts, with a Gene Ontology (GO) pathway enrichment using the Database for Annotation, Visualization, and Integrated Discovery (DAVID) Gene Functional Classification Tool. Although no enrichment pathway was significant after a Bonferroni correction, 14 pathways were enriched prior to correction (p<0.05; modified Fisher's exact test), with the five pathways with the lowest p-values including fluid shear stress and atherosclerosis, relaxin signaling, neuroactive ligand-receptor interaction, HIF-1 signaling, and phosphoinositide-3 kinase (PI3K)-Akt signaling (*Figure 1B*). Many of these pathways have been implicated in various physiological processes, which suggests that variation identified in the hypothalamus are likely the result of autonomic processes in homeostatic maintenance. Individual enrichments of each pathway can be found in *Supplementary file 2*.

We also discovered hundreds of differentially expressed genes (DEGs) between autumn and spring shrews that may mediate phenotypic divergence in both size (shrinkage vs regrowth) and metabolic changes (liver gene expression shifts [*Thomas et al., 2023*]) associated with Dehnel's

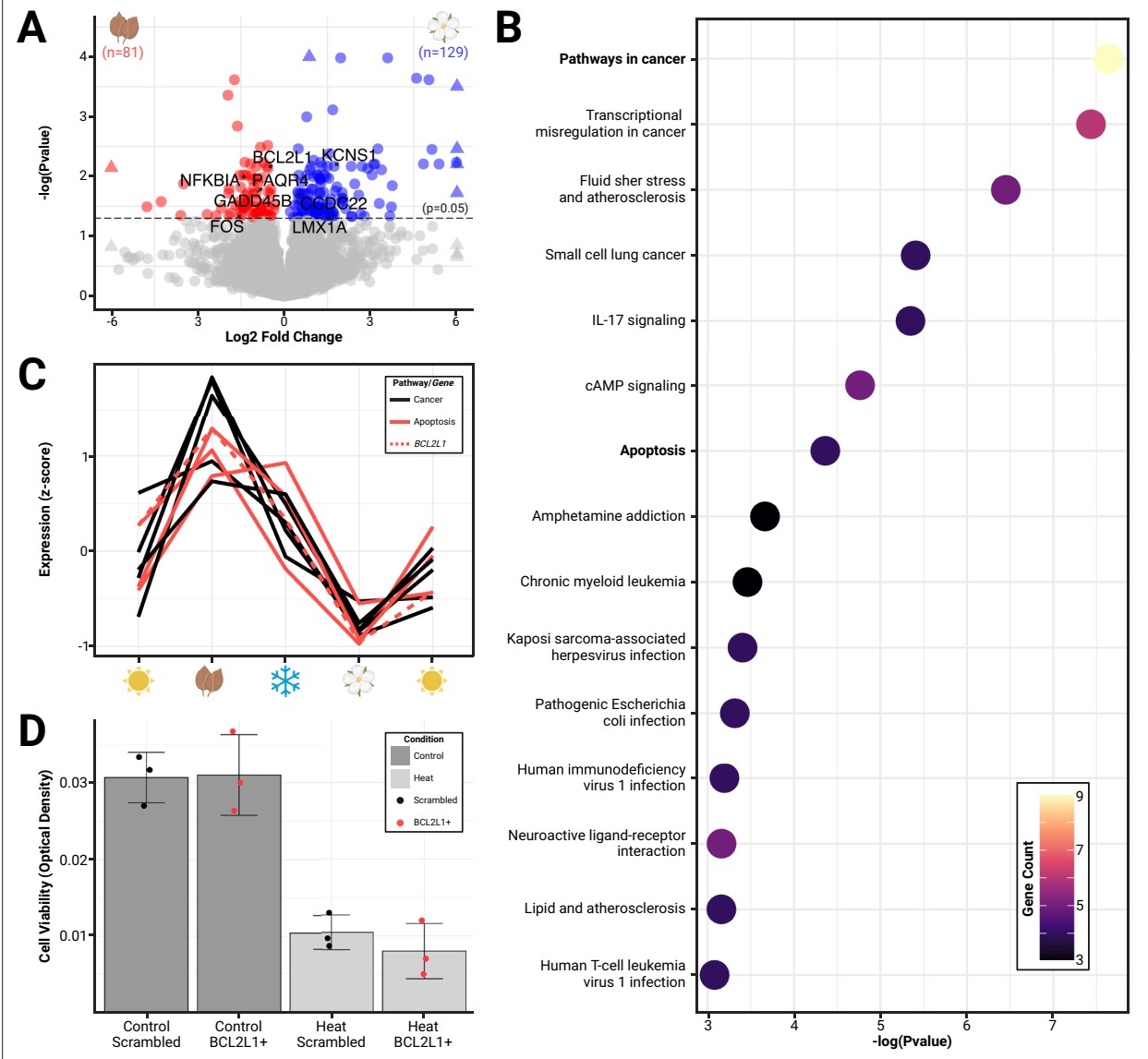

**Figure 2.** Differential gene expression reveals the roles of cellular proliferation and death in seasonal brain size changes. (**A**) Volcano plot of significant (p_adj<0.05) differentially expressed genes (colored) between phenotypic extremes of hypothalami size change (spring vs autumn) plotted by log-fold change. (**B**) Pathway enrichment analysis identified 15 pathways to be enriched for differentially expressed genes, including pathways in cancer and apoptosis (downregulated). (**C**) Patterning of gene expression across seasons of Dehnel's phenomenon for genes found in the cancer (all) and apoptosis (red) pathways, including *BCL2L1* (dashed). (**D**) Cell viability of *Mustela putorius furo* neural cell lines exposed to four treatments: scrambled *BCL2L1* overexpression (n=3), *BCL2L1* overexpression (n=3), heat with scrambled BCL2L1 overexpression (n=3), and heat with *BCL2L1* overexpression (n=3). Error bars represent the mean ± standard deviation. Heat significantly reduced the cell viability compared to controls but was not rescued by *BCL2L1* overexpression.

phenomenon. By comparing the RNA expression of autumn and spring shrews, we found 210 DEGs, with 129 upregulated and 81 downregulated in spring (*Figure 2A*). We ran a GO pathway enrichment to functionally characterize the 210 DEGs in the hypothalamus and found 15 pathways were enriched (p<0.05; modified Fisher's exact test) with significant genes prior to multiple comparison correction. These pathways were all downregulated in spring and included pathways in cancer (4.5-fold enrichment, p<0.001), transcriptional misregulation in cancer (8.3-fold enrichment, p<0.001), fluid shear stress and atherosclerosis (9.5-fold enrichment, p<0.01), small cell lung cancer (11.5-fold enrichment, p<0.01), IL-17 signaling (11.3-fold enrichment, p<0.01), cAMP signaling (5.9-fold enrichment, p<0.01), and apoptosis (7.8-fold enrichment, p<0.05) (*Figure 2B*). Pathways in cancer consisted of nine significantly downregulated genes (from DESeq2 Wald test),

four of which overlapped with apoptotic pathways including *BCL2L1* (–0.6 log-fold change [LFC] $p_{adj}<0.01$), *NFKBIA* (–1.3 LFC, $p_{adj}<0.01$), *FOS* (–1.5 LFC, $p_{adj}<0.05$), and *GADD45B* (–1.4 LFC, $p_{adj}<0.05$) (*Figure 2C*). Differential gene expression for all genes can be found in GitHub supplemental data. Overall, the processes identified suggest regulation of cell proliferation (cancer pathways) and death (apoptosis) in the hypothalamus may be associated with phenotypic changes during Dehnel's phenomenon.

## Cell viability analysis with BCL2L1 overexpression

We examined the potential functional effects of apoptosis regulating gene, *BCL2L1* (*Figure 2D*), which was differentially expressed between seasons, by propagating an in vitro model of domesticated ferret brain cells (MPF-CRL-1656, ATCC). We chose this cell line because there is no established cell line of *S. araneus*. *M. putorius furo* (MPF) also undergoes a Dehnel's-like phenotype (*Apfelbach and Kruska, 1979*) and is more closely related to shrews than mice are. MPF cells were transfected with either the anti-apoptotic *BCL2L1* RNA or a scrambled version of *BCL2L1* (sBCL2L1) RNA. 24 hr after transfection, cells were exposed to heat stress followed by crystal violet staining. A significant difference in cell viability was observed between the heat scrambled compared to the control scrambled groups (df = 3.56, $t_{3.56}$=8.78, p<0.01), however, there was no significant difference between cells transfected with *BCL2L1* compared to those transfected with sBCL2L1 when exposed to heat (df = 3.37, $t_{3.37}$=-0.99, p=0.39).

## Evolutionary divergence in gene expression

Analyses quantifying branch-specific shifts in gene expression identified hundreds of genes that were upregulated in the shrew hypothalamus (post pubescent spring individuals) compared to other mammals, suggesting adaptation for increased expression in these genes (*Figure 3*). Expression variance and evolution (EVE) models found 222 genes significantly rejected ($p_{adj}<0.05$) a single expression optimum for all species in favor of a second expression optimum for the shrew lineage. Notably, all second optima (in the shrew lineage) were higher than the first optimum, consistent with results from the original methods manuscript (*Rohlfs et al., 2015*). While nonsignificant genes had a mean 0.93-fold change, genes experiencing an expression branch-shift in shrews had a mean 9.66 orthologous transcripts per million (oTPM)-fold change compared to other species. To validate genes with expression branch-shifts as shrew-specific, we ran a dropout test for the shrew lineage, testing for phylogeny-wide evidence of selection for each gene. Of the 6496 genes tested, 81 genes had significantly ($p_{adj}<0.05$) lower β (within population variance to between-species variance), suggesting selection elsewhere in the phylogeny. None of these genes overlapped with those identified to be differentially expressed in the shrew, further validating the specificity of expression shifts.

We identified several changes associated with processes that may underlie seasonal size plasticity. Pathway enrichment analysis identified two significantly enriched pathways: calcium signaling (2.8-fold enrichment, p<0.05) and autophagy (9.1-fold enrichment, p<0.05) (*Figure 3*). Upon manual inspection of the list, we also identified several other potential adaptive processes, including BBB formation and function, feeding behavior and leptin responses, metabolism, and neuroprotection (*Figure 3*, *Table 1*).

Many of these putative adaptations mediate responses to environmental cues centered around energy demands. Four of the genes experiencing branch-shift changes in expression were also differentially expressed between autumn and spring individuals, suggesting not only an adaptive shift in the shrew lineage associated with the evolution of Dehnel's phenomenon, but also a direct molecular mechanism for brain changes. The four genes that overlap between both analyses include *Figure 4*: *CCDC22*, which plays an important role in endosomal recycling of membrane proteins (*Singla et al., 2019*); *LMX1A*, a transcription factor essential for dopaminergic neuron development *Hoekstra et al., 2013*; KCNS1, which encodes a subunit of potassium channels associated with neuronal excitability and pain sensitivity *Tsantoulas et al., 2018*; and *PAQR4*, which appears to regulate growth and apoptosis through sphingolipid synthesis in human cancers (*Pedersen et al., 2020*). However, identifying four overlapping genes is not significantly greater than expected by chance (Fisher's exact test, p=0.55).

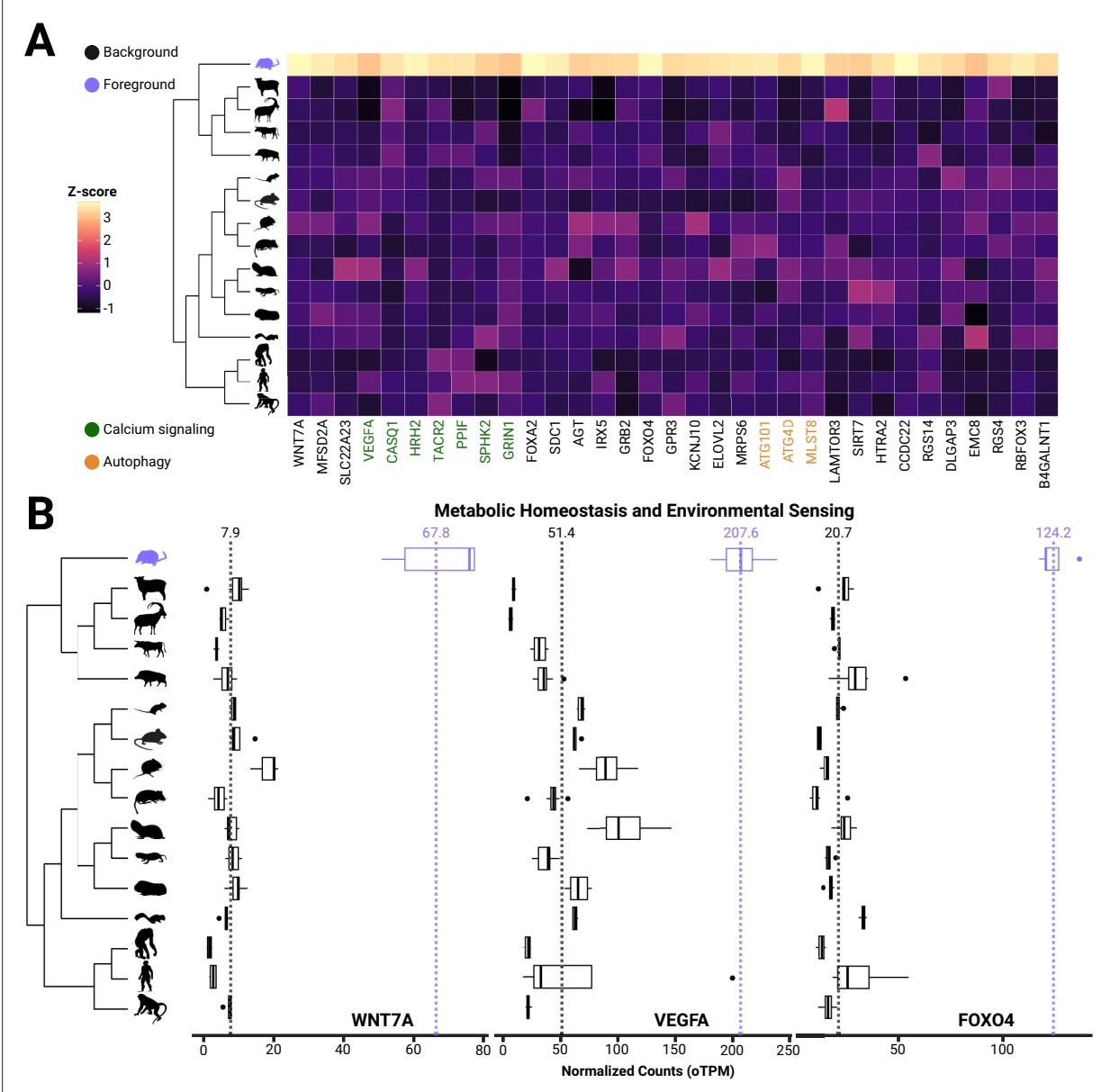

**Figure 3.** Genes involved in blood-brain barrier formation and maintenance are evolutionary upregulated in *S. araneus*. (**A**) Heatmap and boxplots of genes with shrew-specific upregulation compared to other mammals associated with processes including calcium signaling, autophagy, neurological functions, (**B**) metabolic homeostasis and environmental sensing.

The online version of this article includes the following figure supplement(s) for figure 3:

**Figure supplement 1.** Count distribution for each species used in EVE analyses.

## Discussion

By characterizing both seasonal and between-species differential gene expression of the hypothalamus, we generated and probed a unique dataset, discovering expression shifts associated with extreme brain and body size plasticity in *S. araneus*. While the focus on seasonal hypothalamus gene expression provided insights into both the regulation of seasonal homeostasis and processes underlying brain size change, evolutionary shifts in gene expression revealed putative adaptations associated with mammalian brain degeneration and regeneration in a natural system. Our analyses identified a suite of genes related to energy homeostasis and environmental sensing that were either seasonally plastic or upregulated in the shrew compared to other mammals, reinforcing the profound

**Table 1.** Significant shrew-specific upregulation of genes associated with calcium signaling pathways, blood-brain barrier (BBB) plasticity, food intake and leptin response, and other related functions.

| Gene | Function | FC | p-Value | KEGG (p<0.05) |
|---|---|---|---|---|
| WNT7A | BBB/neurogenesis/angiogenesis | 8.60 | 0.000 | NA |
| MFSD2A | Transports DHA across blood-brain barrier/fasting induced | 5.22 | 0.000 | NA |
| SLC22A23 | BBB plasticity | 4.69 | 0.000 | NA |
| VEGFA | BBB plasticity/calcium signaling | 4.04 | 0.024 | Calcium signaling |
| CASQ1 | Calcium signaling/myopathy | 7.22 | 0.000 | Calcium signaling |
| HRH2 | Calcium signaling/sleep related/histaminergic neurons | 11.74 | 0.000 | Calcium signaling |
| TACR2 | Calcium signaling | 14.21 | 0.000 | Calcium signaling |
| PPIF | Calcium signaling | 4.92 | 0.000 | Calcium signaling |
| SPHK2 | Calcium signaling/food intake | 3.18 | 0.025 | Calcium signaling |
| GRIN1 | Calcium signaling/food intake | 3.60 | 0.016 | Calcium signaling |
| FOXA2 | Food intake | 19.37 | 0.000 | NA |
| SDC1 | Food intake | 10.74 | 0.000 | NA |
| AGT | Leptin response/food intake/FOXO1 | 4.23 | 0.029 | NA |
| IRX5 | Leptin response/food intake/neurogenesis | 3.83 | 0.006 | NA |
| GRB2 | Leptin response | 3.36 | 0.003 | NA |
| FOXO4 | Leptin response | 5.99 | 0.000 | NA |
| GPR3 | Thermogenesis/obesity | 3.77 | 0.002 | NA |
| KCNJ10 | Metabolic homeostasis/tanycyte formation | 4.87 | 0.002 | NA |
| ELOVL2 | Lipid metabolism/elongation of VLFA | 6.29 | 0.001 | NA |
| MRPS6 | Mitochondria protein synthesis/Parkinson's | 4.00 | 0.000 | NA |
| ATG101 | Autophagy | 3.95 | 0.000 | Autophagy |
| ATG4D | Autophagy | 3.84 | 0.000 | Autophagy |
| MLST8 | Autophagy/mTor pathway | 8.03 | 0.000 | Autophagy |
| LAMTOR3 | Modulates mTor pathway | 5.37 | 0.000 | NA |
| SIRT7 | Neuroprotective during neurogenesis | 4.64 | 0.004 | NA |
| HTRA2 | Aging/cell and organ size/neuroprotection | 3.86 | 0.000 | NA |
| CCDC22 | NF-KB regulation/Ritscher-Schinzel | 6.95 | 0.000 | NA |
| RGS14 | Suppressed synaptic plasticity (LTP) | 6.21 | 0.002 | NA |
| DLGAP3 | OCD | 5.72 | 0.001 | NA |
| EMC8 | Protein homeostasis of GABAnergic neurons/ER membrane complex | 3.42 | 0.021 | NA |
| RGS4 | GABAnergic/photoperiod/environmental processing | 6.81 | 0.002 | NA |
| RBFOX3 | Promotes sleep/associated with epilepsy | 7.10 | 0.000 | NA |
| B4GALNT1 | Ganglioside synthesis/Promotes BACE1 | 4.30 | 0.008 | NA |

role of metabolism in the evolution and regulation of Dehnel's phenomenon. We also found seasonal changes in genes associated with cancer pathways and apoptosis, indicating that the regulation of cell proliferation and death is critical during brain size change. Notably, some of these results resemble gene expression changes found in human neurological and metabolic disease, which may prove important for future research on therapeutic treatments.

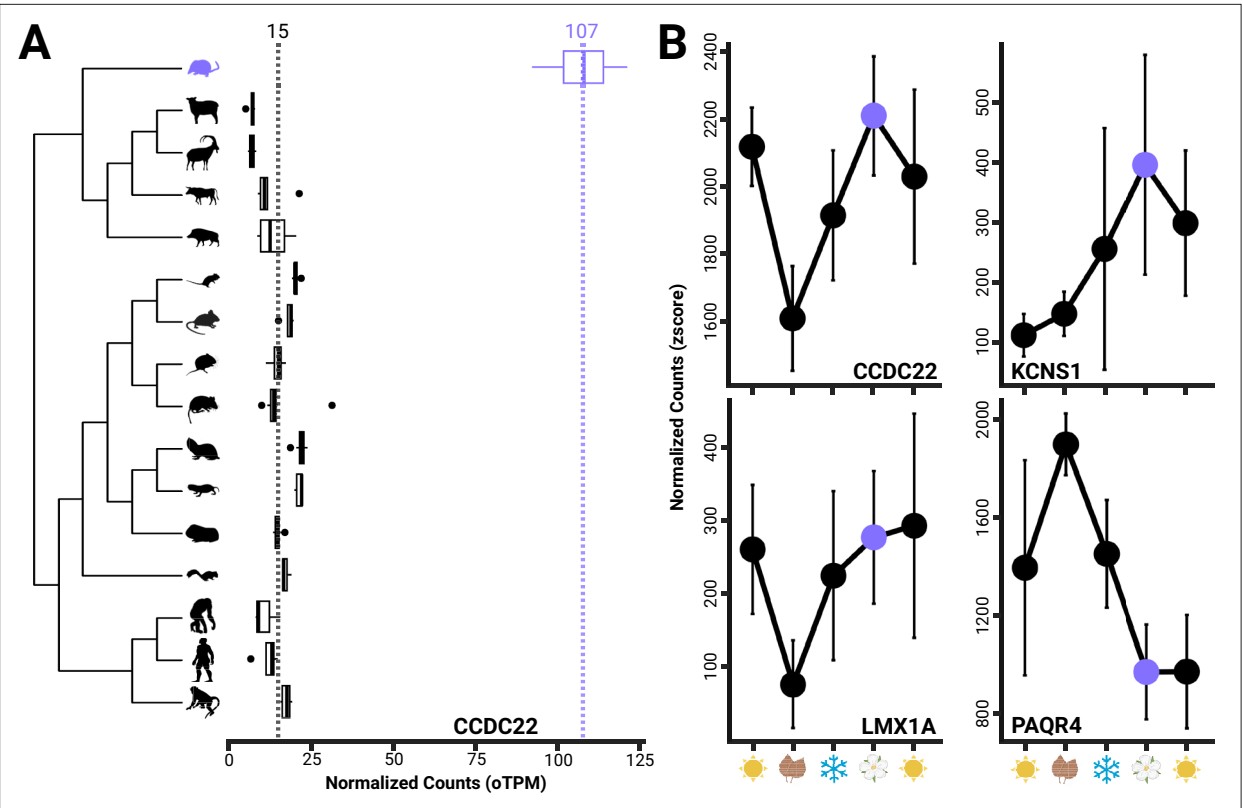

**Figure 4.** Boxplots of CCDC22 showing both evolutionary upregulation (**A**) in the shrew and differential gene expression between spring and autumn individuals, as found in three other genes (*KCNS1*, *LMX1A*, *PAQR4*) (**B**), which have been implicated in the development and progression of human neurological disorders. Error bars represent the mean ± standard deviation per season.

## Seasonal plasticity associated with maintaining metabolic homeostasis

Dehnel's phenomenon has been proposed to permit winter activity by reducing shrew energy requirements by limiting energy devoted to the maintenance of larger tissues (*Pucek, 1970*; *Keicher et al., 2017*; *Taylor, 1998*). This suggests that the maintenance of homeostasis, especially processes related to energy balance, is critical during Dehnel's phenomenon. Seasonal hypothalamus gene expression data corroborate this hypothesis, as we found three pathways enriched with seasonally varying genes in the hypothalamus with metabolic associations: relaxin signaling, PI3K signaling, and neuron ligand-receptor interactions. Relaxin, a neuropeptide, plays a pivotal role in many physiological functions in model organisms, including food and water intake (*McGowan et al., 2009*; *Smith et al., 2011*; *Otsubo et al., 2010*; *McGowan et al., 2005*); food restriction upregulates relaxin-3 (*RLN3*) in vivo (*Lenglos et al., 2013*), while administering RLN3 protein into the rat hypothalamus vastly increases food intake (*de Ávila et al., 2018*; *Calvez et al., 2015*; *Lenglos et al., 2015*; *McGowan et al., 2007*). Relaxin has also been found to activate the PI3K pathway (*Ahmad et al., 2012*; *Dessauer and Nguyen, 2005*), which mediates feeding behavior and glucose homeostasis (*Shen et al., 2011*; *Shen et al., 2017*; *Niswender et al., 2001*; *Könner et al., 2007*; *Hill et al., 2009*), as well as angiogenesis via vascular endothelial growth factor (VEGF) production (*Dessauer and Nguyen, 2005*; *Karar and Maity, 2011*; *Maity et al., 2000*). PI3Ks activity influences feeding behavior and weight gain in response to key metabolic hormones, including leptin and insulin (*Niswender et al., 2001*; *Könner et al., 2007*; *Hill et al., 2009*; *Taniguchi et al., 2006*). Lastly, the neuroactive ligand-receptor interaction pathway, a subclass of Kyoto Encyclopedia of Genes and Genomes (KEGG) Environmental Information Processing pathway, is associated with responses in model organisms to shifts in diet (*Li et al., 2018*), stress (*Chen et al., 2021*), and temperature (*Kim et al., 2017*; *Chen et al., 2014*), all of which the shrew experiences during seasonal changes. Although exploration of gene function in *S. araneus* has only begun, seasonal gene expression in the hypothalamus highlights that while remaining active

in winter with high metabolic rates (*Taylor, 1998*), shrews actively regulate feeding behavior, food intake, and energy homeostasis.

## Putative hypothalamus adaptations may enhance regulatory responses

*S. araneus* have disproportionately high metabolic rates (*Taylor, 1998*), which they must maintain in winter. While seasonal transcriptomics indicate active regulation of processes related to energy homeostasis, we hypothesized that natural selection may select for effective strategies in comparison to other mammals. To test this, we used comparative analyses which indicated shrews may have evolved adaptive environmental sensing. We found upregulation of several proangiogenic genes associated with the formation and function of the BBB, including *VEGFA* and *WNT7A*. In normal neural development, neural progenitor cells and surrounding astro- and pericytes express these two genes defining an expression gradient that guides blood vessels to form the BBB (*Raab et al., 2004*; *Carmeliet et al., 1996*; *Daneman et al., 2009*).

While this suggests *VEGFA* and *WNT7A* upregulation in the shrew may increase vascularization of the brain, continued expression of *VEGFA* beyond development increases BBB permeability, leading to a breakdown of its protective effects (*Argaw et al., 2012*; *Kim et al., 2008*). BBB permeability is a common symptom of many disease-like states including neuroinflammation (*Argaw et al., 2009*) and neurodegeneration (*Argaw et al., 2009*; *Lan et al., 2022*; *Obermeier et al., 2013*). The hypothalamus BBB consists of tanycytes and highly fenestrated capillaries with less tight junctions (*Bennett et al., 2009*), replacing the less permeable BBB of other brain regions with a specialized barrier that improves nutrient and energy sensing through dynamic passage of molecules from peripheral circulation (*Langlet et al., 2013*; *Haddad-Tóvolli et al., 2017*; *Lam et al., 2005*). Experiments on mice indicate reduced blood glucose levels from fasting promote capillary fenestration in the hypothalamic BBB via upregulation of *VEGFA*, eliciting feeding responses through increased hypothalamic exposure to glucose levels (*Langlet et al., 2013*). In shrews, upregulation of *VEGFA* and *WNT7A* may constitutively increase capillary fenestration, a potential adaptation of the BBB to improve hypothalamus metabolic sensing and signaling in response to high energy demands. Notably, adaptive expression of *VEGFA* has also been identified in the seasonal vascular plasticity of other mammal species, including reproductive sheep (*Chevillard et al., 2022*; *Castle-Miller et al., 2017*), bison (*Tabecka-Lonczynska et al., 2018*), and squirrels (*Yao et al., 2021*), thus shrew upregulation of *VEGFA* to meet unusually high metabolic demands may parallel mechanisms used seasonally in reproduction.

We also found evidence for adaptive gene expression relating to maintenance of energy balance, specifically in feeding behavior and leptin responses. Key genes associated with the hypothalamic leptin response, such as *NPY*, *AgRP*, or *POMC*, were not universally orthologous and so could not be analyzed, but there was evolutionary upregulation of many genes involved in downstream leptin responses (*FOXO4*, *GRB2*, *IRX5*, *AGT*) (*Figure 3*, *Table 1*). Unlike their upstream effectors, these genes may have subtle effects on continuous feeding behavior. For example, in the hypothalamus FOXO1 negatively regulates POMC and promotes AgRP expression (*Ma et al., 2015*), which can contribute to leptin resistance resulting in a long-term appetite suppression (*Amitani et al., 2013*; *Sasaki et al., 2010*; *Nakae et al., 2006*). However, much less is known about the hypothalamic functions of the *FOXO1* paralog, *FOXO4*. In mouse adipocytes, leptin-dependent expression of *FOXO4* can rapidly clear blood glucose levels (*Wang et al., 2009*), which can lead to energy storage and decreased satiety. We hypothesize that, in shrews, upregulation of *FOXO4* could either increase blood glucose clearance or leptin resistance, reducing long-term satiation and promoting continuous feeding. Functionally, evolutionary upregulation of these genes in the shrew hypothalamus may result in evolved leptin insensitivity, instead of the development of leptin resistance. Reduced leptin sensitivity in shrews could reduce overall satiation, promoting foraging even when the short-term energy budget is balanced thereby improving anticipatory storage before winter.

Lastly, calcium signaling is a ubiquitous form of neural communication (*Ahrens et al., 2013*; *Mann et al., 2017*) that occurs through the regulation of intra- and extracellular calcium concentrations, transmitting signals that can release neurotransmitters (*Kater et al., 1988*; *Mattson, 1988*) and promote gene expression (*Deisseroth et al., 1996*; *Bito et al., 1997*; *Berridge, 1998*; *Zhang et al., 2009*). Thus, it has been implicated in numerous functions including dendrite growth (*Redmond and Ghosh, 2005*; *Lohmann and Wong, 2005*), synaptogenesis (*Kazama et al., 2007*; *Verderio et al., 1994*; *Michaelsen and Lohmann, 2010*), and initiation and maintenance of long-term potentiation (*Malenka*

et al., 1988; Poser and Storm, 2001). In shrews, adaptive calcium signaling through the upregulation of several calcium-responsive genes (VEGFA, CASQ1, HRH2, TACR2, PPIF, SPHK2, GRIN1) may improve environmental information relay in the hypothalamus, both internally and throughout the CNS.

## Evolution of brain size plasticity: roles of cellular proliferation, death, and links to neurological disease

Progressive cell loss in neurons is a hallmark of neurodegeneration (Wilson et al., 2023), but despite seasonal decreases in brain mass and volume, shrews show no reduction in cell number (Smith, 2016). Uncorrelated changes in size and neuron count in shrews might suggest they evolved neuroprotective mechanisms to reduce cell loss associated with decreased brain size. We found an enrichment of genes upregulated in autumn shrews involved in cancer pathways and apoptosis regulation. The processes underlying cancer and neurodegeneration oppose one another, with increased resistance to cell death driving cancer and premature cell death driving neurodegeneration. Shrews appear to traverse this tightrope by regulating these pathways to avoid both the cell loss typical of neurodegeneration and unchecked proliferation of cancer. This may represent a unique neuroprotective mechanism.

BCL2L1-mediated apoptosis seems to play a critical role in helping shrews maintain this balance during seasonal size change. As a known anti-apoptotic gene, BCL2L1 promotes cell survival by reducing apoptosis through an NFKB-dependent pathway, as demonstrated in rodents (Bui et al., 2001; Motoyama et al., 1995; Boise and Thompson, 1997). The BCL2 gene family is also recognized for its contrasting roles in disease, with upregulation in CNS disorders and downregulation in cancers. We found an enrichment of the apoptosis pathway with upregulated genes in autumn, including BCL2L1, which may explain why shrews do not exhibit the neuronal loss typically associated with human neurodegeneration. This suggests that the mechanism parallels aspects of neurodegenerative disease without the associated pathology, which we propose could be another neuroprotective mechanism to reduce cell death during brain shrinkage in shrews.

The regulation of apoptosis may represent a convergently evolved strategy across species managing environmental and metabolic stress. For example, natural experiments in hibernators found seasonal reduction of apoptosis, paired with upregulation of BCL2L1 (Fleck and Carey, 2005; Logan et al., 2016), including in the brain (Rouble et al., 2013). Specific to Dehnel's phenomenon, BCL2L1 upregulation in autumn indicates tight regulation of cell survival during brain size changes and periods of metabolic and environmental stress: a balance between brain shrinkage and maintenance of cell numbers.

Evolutionary change in shrew gene expression likely benefits this cellular regulation. We found an evolutionary upregulation of genes associated with autophagy, which is the degradation of cellular parts to provide energy and materials to maintain cellular homeostasis under stressful conditions, in the shrew hypothalamus. This included MLST8, an integral component of the mTOR protein complex 1 (mTORC1) that regulates sugar and lipid metabolism (Guertin et al., 2006), but also has recently been found to modulate autophagy in response to nutrient deprivation (Saxton and Sabatini, 2017). Two evolutionarily upregulated genes PPIF and SPHK2 can also promote apoptosis associated with mitochondrial calcium influx. PPIF is an important component of the mitochondrial permeability transition, which responds to cellular calcium overload and oxidative stress by promoting cell death (Baines et al., 2005; Nakagawa et al., 2005; Li et al., 2009; Halestrap, 2004). Similar to PPIF, overexpression of SPHK2 in stressed cells induces cell cycle arrest and promotes apoptosis through a series of signaling cascades that increase cytosolic calcium and promote the release of pro-apoptotic cytochrome c from the mitochondria (Maceyka et al., 2005; Liu et al., 2003). While both SPHK2 and PPIF are overexpressed and likely contribute to neuronal death in both human patients with Huntington's disease or Alzheimer's disease and rodent models of these diseases, knockout or inhibition of these genes protects against neurodegeneration (Moruno-Manchon et al., 2017; Takasugi et al., 2011; Du et al., 2008; Du et al., 2011). In the shrew hypothalamus, canalized upregulation of these genes likely influences cell survival and death, but precisely how requires further investigation.

Lastly, CCDC22, KCNS1, LMX1A, and PAQR4 are both seasonally plastic and evolutionarily upregulated in the shrew hypothalamus (Figure 4). While the number of overlapping genes is not significantly greater than expected by chance, the combination of seasonal and evolutionary upregulation suggests these genes may play integral roles in Dehnel's phenomenon. Notably, some of these genes

are directly related to the development and progression of neurological disorders in humans. For example, CCDC22 regulation in shrews may provide protection against harm incurred during seasonal size changes. The coiled-coil domain containing 22 protein (CCDC22) plays a key role in endosomal recycling of proteins and is a novel candidate gene for several neurological disorders (*Voineagu et al., 2012*; *Kolanczyk et al., 2015*; *Neri et al., 2022*; *Gjerulfsen et al., 2021*). CCDC22 is a subunit of the CCC (CCDC22, CCDC93, COMMD) complex that aids in trafficking and recycling of endosomal membrane proteins (*Singla et al., 2019*; *Phillips-Krawczak et al., 2015*; *Bartuzi et al., 2016*). Improper recycling and resulting protein aggregation readily occurs in neurons, as they do not proliferate (*Rubinsztein, 2006*; *McDonald, 2021*), and has also been implicated in neurodegenerative diseases, including cytotoxicity in amyotrophic lateral sclerosis (*Muzio et al., 2020*), amyloid-B aggregation in Alzheimer's disease (*Small et al., 2005*; *Rogaeva et al., 2007*; *Lane et al., 2010*), and MUL1 degradation in Parkinson's disease (*Tang et al., 2015*). Missense mutations and subsequent downregulation of CCDC22 are potential mechanisms underlying X-linked intellectual disability (*Voineagu et al., 2012*; *Kolanczyk et al., 2015*) and Ritscher-Schinzel syndrome (*Kolanczyk et al., 2015*; *Neri et al., 2022*; *Gjerulfsen et al., 2021*) in humans. Both disorders are characterized by macrocephaly, malformations of the brain, craniofacial abnormalities, and intellectual disabilities. In pubescent spring shrews, we found CCDC22 has an ~7-fold upregulation compared to other mammals and a significant upregulation during spring compared to autumn shrinkage. Neuroprotective functions may be especially important in spring, as brain mass returns increase the need for endosomal recycling of proteins and may explain seasonal plasticity in *CCDC22* expression in tandem with adaptive upregulation.

## Limitations

Despite the proposed adaptive roles of the many genes discussed in our study, several limitations must be acknowledged. During shrew seasonal size plasticity, shrew environments may differ from those of the other species included in this experiment. These environmental differences could influence gene expression, leading to over- or underestimation of selection in the shrew lineage. Although we controlled for several sources of variation by using only pubescent and wild or non-domesticated individuals when feasible, other sources of variation remain unaccounted for because of practical constraints. The functions we propose for many genes remain speculative and require further validation. For example, we tested the hypothesis that autumn upregulation of *BCL2L1* reduces apoptosis by overexpressing this gene in a related mammalian neural cell line. However, this experiment did not demonstrate a rescue effect against heat-induced apoptosis, contrary to our expectations. While this unexpected result could stem from several causes, such as the anti-apoptotic function of *BCL2L1* being specific to the unique neural environment of shrews or arise through species-specific gene interactions, it may also reflect limitations in our study, highlighting the need for further investigation to validate and refine our conclusions.

## Conclusion

We discovered evidence for adaptive evolution of gene expression involved in the regulation of metabolism, cell survival, and size plasticity associated with Dehnel's phenomenon. Not only does the hypothalamus shrink to reduce energetic load during winter, but it must also control metabolic homeostasis during these changes. Our results reveal both conserved and novel gene expression mechanisms that likely underlie the central functions of the hypothalamus during seasonal size plasticity. While previous work had hinted at the metabolic nature of Dehnel's phenomenon, by pairing seasonal differential gene expression with comparisons to other species, we identified adaptations likely involved in the evolution of this unique brain size plasticity. Metabolism, body size, and longevity are intrinsically linked life history factors. In the shrew, unusually high metabolic rates force winter activity, leading to intense metabolic sensing and regulation, changes in cell survival pathways, and—through both the evolution of Dehnel's phenomenon and these sensing and regulatory adaptations—shortened lifespans. Together, these gene expression adaptations likely underlie drastic seasonal size change and mitigate its detrimental effects.

## Materials and methods

### Shrew sample collection

We used tissues from shrews collected for experiments analyzing gene expression and metabolic shifts in brain regions through Dehnel's phenomenon (*Thomas et al., 2023*). Briefly, S. *araneus* were collected from a single German population (47.9684 N, 8.9761 E) across five different seasons of Dehnel's phenomenon from June 2020 to June 2021 (n=25, $n_{per\_season}$ = 4–5, protocols authorized by Regierungspräsidium Freiburg, Baden-Württemberg 35-9185.81/G-19/131). Shrews were trapped with insulated wooden traps that contained mealworms, which were checked every 2 hr. This protocol was used to minimize trap-related stress from heat/cold shock or lack of food and thus reduce stress-related variation on gene expression. Shrews were then euthanized via vascular perfusion of PAXgene Tissue Fixative. Brain regions were dissected into individual regions in cold fixative and then incubated in PAXgene Tissue Stabilizer for 2–24 hr before long-term storage. Samples were preserved in stabilizer at –180°C in liquid nitrogen until RNA extraction.

### RNA extraction, library preparation, and sequencing

We extracted the *S. araneus* hypothalamus RNA using a modified QIAGEN Micro RNeasy protocol used for small mammalian sensory tissues (*Yohe et al., 2020*). These extractions reduce degradation due to heat and improve RIN from standard QIAGEN Micro RNeasy protocols by disrupting tissue on dry ice using glass, instead of plastic, mortar, and pestles. RNA was sent to Azenta Life Sciences for quality control (nanodrop and RNA ScreenTape), library preparation, and sequencing. Hypothalamus libraries were prepared with poly-A selection and sequenced for approximately 15–25 million reads per sample in 150 bp PE reads.

RNA sequences from other species were collected from the NIH National Center for Biotechnology Information's (NCBI) Sequence Read Archive datasets (*Supplementary file 1*). First, we searched the Sequencing Read Archives (SRA) for the keyword hypothalamus, filtered by both RNA and mammal species. As models described below rely on using the variance both between and within species, species required three or more biological samples to be included. This removed any species that had only one hypothalamus sample sequenced. Species were excluded from the study if a genome assembly was not readily available (e.g. *Phodopus sungorus*), as this information was needed for an unbiased alignment of reads. Our final criterion was to only include species for which post-pubescent individuals were available to reduce noise from age effects and the onset of puberty. If the onset of puberty was not specifically stated for the dataset, AnAge (*Tacutu et al., 2018*) was used to determine if samples were pubescent. *Oryctolagus cuniculus* and *Pan paniscus* were prepubescent and thus removed. For the remaining species, if multiple hypothalamus RNA-seq datasets were available, we used less stringent rules to determine which data to select. First, to reduce domestication or captivity effects, we chose wild or non-domesticated individuals over laboratory-raised animals. Second, to reduce the variance from different extraction protocols and sequencing methods, we used samples from the same experiments or larger datasets, available for multiple species or brain regions. Using samples from datasets with multiple regions or species also suggests intimate knowledge of neural anatomy, increasing confidence in dissections. Third, we used a maximum of eight samples per species and attempted to maintain a 1:1 sex ratio when possible. We did not filter by sequencing method (e.g. used 50 bp SE as well as 150 PE reads) or sequencing depth, as filtering poor reads and normalizing for library depth and content should account for both these factors.

### RNA quantification, normalization, and orthology

Adapter sequences were trimmed from the raw reads using the default parameters of fastp (*Chen et al., 2018*), which is able to autodetect adapters regardless of different library preparations across species. This program also corrected and pruned low-quality bases and reads using a sliding window approach. Processed reads were aligned and quantified with Kallisto 0.46.2 (*Bray et al., 2016*), which probabilistically estimates gene counts through pseudo-alignment to each species-specific genome assembly. Samples were removed from the dataset if mapping rate was below 30%, indicating either poor sequencing or low assembly quality. Additionally, novel shrew hypothalamus sequences were compared against previously published shrew region data to verify tissue type. Gene counts for all species were then normalized for total library size into TPM. Orthologous genes between species were inferred with OrthoFinder (*Emms and Kelly, 2019*) using default parameters, retaining only

single-copy orthologs identified across all species (n=6496 single-copy orthologs). Visualizations of the frequency distribution of the oTPMs (*Figure 3—figure supplement 1*) were used to identify outlier species. For example, *Papio anubis* was removed from this experiment because of its extremely low mapping percentage (mean 16.75%; range 9.5–23.7%), which had a large effect on the oTPM distribution.

## Branch-shift changes in gene expression

We modeled gene expression as a trait using phylogenetic comparative methods, as these account for evolutionary divergence among species (*Dunn et al., 2018*). We ran EVE models to test for significant changes of gene expression in the shrew lineage, using post-pubescent spring individuals (*Rohlfs et al., 2015*). EVE models both parameterize and estimate the ratio (β) of population (within species) to evolutionary (between species) variance, such that high β ratios indicate gene expression plasticity, and low β ratios indicate differential expression between species, necessary for inferring selection. We ran two EVE models to test for divergence in gene expression in the shrew lineage. First, we tested for branch-specific gene expression level shifts by contrasting the likelihood of two Ornstein-Uhlenbeck models for the data, one with a single expression optimum (null; stabilizing selection) and another with a second optimum on the shrew branch (selection). A likelihood ratio test between the null and selective hypotheses for each gene ($\chi_1^2$) was conducted using these likelihoods. A Bonferroni correction was used to account for multiple hypothesis tests to identify candidate genes under selection ($p_{adj}<0.05$). For the second model, we ran a dropout test to further validate the specificity of the gene expression change in the shrew lineage. After removing the shrew gene expression data, we identified genes with high expression divergence across the phylogeny (significantly low β ratios), which would indicate relaxed or diversifying selection. This gene list was compared to that from first model to prune genes not specifically associated with divergence in the shrew. Both models described above ran on the Bayesian molecular-clock mammalian phylogeny of *Álvarez-Carretero et al., 2022*, pruned to match our species sample. Finally, we used the DAVID functional annotation tool (*Huang et al., 2009*) to determine enriched KEGG GO pathways from the candidate gene set.

## Temporal clustering

To analyze the temporal variation of shrew gene expression in the hypothalamus, we temporally clustered our data using the package TCseq (*Mengjun, 2019*). We began by further normalizing our data across Dehnel's phenomenon seasons using the median of ratios from DESeq2 (*Love et al., 2014*), which normalized for library size and content. Genes with consistent expression across time with little variation between seasons would not be associated with observed phenotypic changes. These genes were filtered from this dataset by only selecting genes with an absolute fold change of 0.5 between any two seasons. We also removed genes with low gene expression that would appear to have high fold changes despite lacking enough transcript expression to influence phenotype, retaining only reads with two samples >10 normalized reads. After filtering, counts for each season were converted to mean z-scores and then clustered into groups of similar gene expression profiles using fuzzy c means clustering. The number of resulting gene clusters was calculated a priori, by bootstrapping (n=20) a gap statistic that minimizes within-cluster distances. Genes with highest membership in clusters associated with variation across all seasons (seasonal effect; Clusters 1, 4, 5, 6, 7, 9, 10), as compared to those that just show extensive differences (>1.25 standard deviations) between the summer seasons (juvenile and adult) without a second local maximum (abs(autumn-winter)<0.5 and abs(winter-summer adult)<0.5) (developmental effect; Clusters 2, 3, 8, 11, 12), were analyzed for KEGG GO enrichment using DAVID functional annotation (*Huang et al., 2009*). Additionally, each cluster was analyzed individually and included in *Supplementary file 2*.

## Differential gene expression

We tested for differential gene expression between autumn and spring individuals, as these seasons differ in hypothalamic size phenotype (shrinkage vs regrowth) (*Lázaro et al., 2018*) and were previously identified divergence in liver gene expression related to metabolism (*Thomas et al., 2023*). To test for significant differential gene expression, we fit a negative binomial generalized linear model to the normalized (median of ratios) gene counts using DESeq2 (*Love et al., 2014*). We then tested for significant differences in gene expression between autumn and spring using a Wald test, followed by

multiple testing correction on resulting p-values with the Benjamini and Hochberg procedure (*Benjamini and Hochberg, 1995*). Sex was used as an additional covariate (~sex + condition), but our design is imbalanced as not every season samples both sexes. Significantly DEGs were binned by fold change (>1.58 LFC), to quantify DEGs of high effect. Significant DEGs were also used to identify KEGG GO enrichment (*Huang et al., 2009*), and compared against the candidate list of genes with branch-shift changes in expression to identify genes both plastic across seasons and consistent with gene expression adaptation in the shrew. To test if observed overlap was greater than expected by chance, we performed a Fisher's exact test.

## Cell culture

MPF brain cells were obtained and authenticated (CO1 assay) by ATCC (CRL-1656, LOT 70047219, ampule passage number 111). No mycoplasma contamination was detected, tested with Hoechst DNA stain, agar culture, and PCR-based assays. These cells were cultured in 10 mL of Basal Medium Eagle supplemented with 14% sheep serum. Cells were then cultured in 60.8 cm² treated tissue culture dishes at 37°C in 5% $CO_2$ atmosphere and 95% humidity. With induced cell death, cells were seeded into treated flat bottom six-well plates with 2 mL/well of complete culture media.

## Cell death induction

Three experiments were conducted to induce cell death in MPF cells (cold induced, peroxide induced, and heat induced), with only heat causing significant decrease in cell viability. *Cold induced*: Two 60.8 cm² tissue culture dishes of MPF cells were seeded into two six-well plates for 24 hr before reaching 80% confluency. In one experiment, 'short cold' apoptosis was performed by immersing the six-well plate into a 0°C ice-water bath for 2 hr, followed by 3 hr of rewarming to 37°C in an incubator. Cells were then treated with crystal violet staining. In a similar experiment, 'long cold' apoptosis was performed by immersing the six-well plate into a 0°C water bath for 4 hr, followed by 24 hr of rewarming to 37°C before crystal violet staining. *Heat-induced cell death*: Two 60.8 cm² tissue culture dishes of MPF cells were seeded into two six-well plates for 24 hr before reaching 80% confluency. Heat cell death was attempted by immersing the six-well plate into a 45°C water bath for 2 hr, followed by re-cooling to 37°C prior to crystal violet staining.

## In vitro transcription and transfection of cells

The AmpliScribe T7 High Yield Transcription kit (Lucigen) was used for RNA in vitro transcription with a 2 hr incubation at 37°C using gblock sequences available in supplements. BCL2L1 sequences were designed with a T7 promoter inserted at the beginning to facilitate transcription (Integrated DNA Technologies). We selected *BCL2L1* because it is the gene furthest downstream in the apoptotic pathway, thus making it the most directly involved gene in programmed cell death, whereas upstream genes could influence additional genes or alternative processes. RNA transcripts were purified using the QIAGEN miRNA kit and concentration determined using a NanoDrop 2000 (Thermo Fisher Scientific). Two 60.8 cm² tissue culture dishes of MPF cells were seeded into two six-well plates for 24 hr before reaching 80% confluency. 300 ng of *BCL2L1* RNA or a scrambled form of *BCL2L1* RNA were transfected with Lipofectamine 3000 into each well for 24 hr according to the manufacturer's instructions (Thermo Fisher Scientific). In short, two tubes were used to make a master mix for each RNA sample. Tube 1 contained 125 µL/well of Opti-MEM (Thermo Fisher Scientific) with 7.5 µL/well of Lipofectamine 3000. Tube 2 contained 125 µL/well of Opti-MEM, 5 µL/well of P3000, and 300 ng/well of RNA. Tube 2 was added to tube 1 and incubated 15 min at room temperature. The solution was then gently mixed and dispersed evenly to cells.

## Crystal violet cell viability assays

Crystal violet solution was made using 500 mg crystal violet powder in 100 mL of 50% methanol. Cells were cultured to 80% confluency in two six-well plates. Complete media was aspirated in wells and washed three times in 1 mL of 1× DPBS before 500 µL of crystal violet solution was added to each well. Plates were wrapped in foil and placed on a shaker for 30 min. After time elapsed, crystal violet was removed, and cells were washed with tap water until free color was no longer visible. Plates were left at room temperature for 10–15 min until dry. 500 µL of 100% methanol was added to each well and plates were put on a shaker for 1 hr at room temperature. 100 µL of solution was taken from

each well in triplicate and added to a 96-well plate. 100 μL of 100% methanol was added in triplicate to a row for normalization of background absorbance. Wells were read at 570 nm using a spectrophotometer (BioTek). For viability, data are reported as a percentage of control or optical density for absorbance. For statistical analysis, differences were judged to be statistically significant when $p < 0.05$ by a Welch's two-sample t-test.

## Acknowledgements

Human Frontiers of Science Program, award: RGP0013/2019 (to Dina KN Dechmann, John Nieland, Liliana M Dávalos). WRT was supported in part by a Stony Brook University Presidential Innovation and Excellence award to LMD.

## Additional information

### Funding

| Funder | Grant reference number | Author |
| --- | --- | --- |
| Human Frontier Science Program | 10.52044/HFSP. RGP00132019.pc.gr.164597 | John D Nieland Dina Dechmann Liliana M Davalos |
| Stony Brook University | Presidential Innovation and Excellence Award | Liliana M Davalos |

The funders had no role in study design, data collection and interpretation, or the decision to submit the work for publication.

### Author contributions

William R Thomas, Conceptualization, Data curation, Software, Formal analysis, Investigation, Visualization, Methodology, Writing – original draft, Writing – review and editing; Troy Richter, Resources, Formal analysis, Investigation, Visualization, Methodology, Writing – original draft, Writing – review and editing; Erin T O'Neil, Resources, Formal analysis, Validation, Investigation, Visualization, Methodology, Writing – original draft, Writing – review and editing; Cecilia Baldoni, Resources, Investigation, Writing – review and editing; Angelique Corthals, Funding acquisition, Validation, Writing – review and editing; Dominik von Elverfeldt, Writing – review and editing; John D Nieland, Funding acquisition, Writing – review and editing; Dina Dechmann, Conceptualization, Resources, Funding acquisition, Project administration, Writing – review and editing; Richard Hunter, Conceptualization, Resources, Supervision, Methodology; Liliana M Davalos, Funding acquisition, Methodology, Project administration, Writing – review and editing

### Author ORCIDs

William R Thomas ⓘ https://orcid.org/0009-0002-3858-2440
Troy Richter ⓘ https://orcid.org/0000-0003-4142-7089
Erin T O'Neil ⓘ https://orcid.org/0009-0008-9307-6915
Cecilia Baldoni ⓘ https://orcid.org/0009-0008-1341-9456
Dominik von Elverfeldt ⓘ https://orcid.org/0000-0002-6219-3528
John D Nieland ⓘ https://orcid.org/0000-0001-7423-0122
Dina Dechmann ⓘ https://orcid.org/0000-0003-0043-8267
Liliana M Davalos ⓘ https://orcid.org/0000-0002-4327-7697

### Ethics

Eurasian common shrews, Sorex aranaues, were trapped and euthanized according to protocols authorized by Regierungspräsidium Freiburg, Baden-Württemberg (35-9185.81/G-19/131) in Radolfzell, Germany (47.9684N, 8.9761 E).

Reviewer #1 (Public review): https://doi.org/10.7554/eLife.100788.3.sa1
Reviewer #2 (Public review): https://doi.org/10.7554/eLife.100788.3.sa2
Reviewer #3 (Public review): https://doi.org/10.7554/eLife.100788.3.sa3

Author response https://doi.org/10.7554/eLife.100788.3.sa4

## Additional files

### Supplementary files

Supplementary file 1. RNA sequencing information from species collected from the NIH National Center for Biotechnology Information's (NCBI) Sequence Read Archive datasets.

Supplementary file 2. Kyoto Encyclopedia of Genes and Genomes (KEGG) Gene Ontology (GO) enrichment using Database for Annotation, Visualization, and Integrated Discovery (DAVID) functional annotation for each cluster identified in TCseq.

MDAR checklist

### Data availability

Supplementary tables, data, results, and code deposited and found on Github https://github.com/wrthomas315/Sorex_Hypothalamus_Transcriptomics2 (copy archived at *Thomas, 2025*). Raw sequencing data located in the NCBI Sequencing Read Archive (BioProject PRJNA941271).

The following dataset was generated:

| Author(s) | Year | Dataset title | Dataset URL | Database and Identifier |
|---|---|---|---|---|
| Thomas WR, Richter T, O'Neil ET, Baldoni C, Corthals AP, von E, Nieland J, Dechmann DKN, Hunter RG, Dávalos LM | 2023 | Shrinking Shrews: molecular mechanisms of seasonal size change | https://www.ncbi.nlm.nih.gov/bioproject/PRJNA941271 | NCBI BioProject, PRJNA941271 |

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
