## [Editor Report · eLife Assessment]

This study presents **valuable** findings related to seasonal brain size plasticity in the Eurasian common shrew (Sorex araneus), which is an excellent model system for these studies. The evidence supporting the authors' claims is **convincing**. The work will be of interest to biologists working on neuroscience, plasticity, and evolution.

---

## [Referee Report · Reviewer #1 (Public review)]

Summary:

In this paper, Thomas et al. set out to study seasonal brain gene expression changes in the Eurasian common shrew. This mammalian species is unusual in that it does not hibernate or migrate but instead stays active all winter while shrinking and then regrowing its brain and other organs. The authors previously examined gene expression changes in two brain regions and the liver. Here, they added data from the hypothalamus, a brain region involved in the regulation of metabolism and homeostasis. The specific goals were to identify genes and gene groups that change expression with the seasons and to identify genes with unusual expression compared to other mammalian species. The reason for this second goal is that genes that change with the season could be due to plastic gene regulation, where the organism simply reacts to environmental change using processes available to all mammals. Such changes are not necessarily indicative of adaptation in the shrew. However, if the same genes are also expression outliers compared to other species that do not show this overwintering strategy, it is more likely that they reflect adaptive changes that contribute to the shrew's unique traits.

The authors succeeded in implementing their experimental design and identified significant genes in each of their specific goals. There was an overlap between these gene lists. The authors provide extensive discussion of the genes they found.

The scope of this paper is quite narrow, as it adds gene expression data for only one additional tissue compared to the authors' previous work in a 2023 preprint. The two papers even use the same animals, which had been collected for that earlier work. As a consequence, the current paper is limited in the results it can present. This is somewhat compensated by an expansive interpretation of the results in the discussion section, but I felt that much of this was too speculative. More importantly, there are several limitations to the design, making it hard to draw stronger conclusions from the data. The main contribution of this work lies in the generated data and the formulation of hypotheses to be tested by future work.

Strengths:

The unique biological model system under study is fascinating. The data were collected in a technically sound manner, and the analyses were done well. The paper is overall very clear, well-written, and easy to follow. It does a thorough job of exploring patterns and enrichments in the various gene sets that are identified.

I specifically applaud the authors for doing a functional follow-up experiment on one of the differentially expressed genes (BCL2L1), even if the results did not support the hypothesis. It is important to report experiments like this and it is terrific to see it done here.

Comments on revised version:

This updated version of the paper is improved compared to its initial version. As such, the strengths remain the same as before, with a fascinating model system and an interesting research question. The earlier weaknesses related to overinterpretation of the data have been largely fixed by shortening the paper and adding appropriate caveats throughout. The paper now also includes a significance test for its overlap between gene lists. While this turned out to be negative (i.e., there is not more overlap between lists than expected by chance), reporting this result transparently has strengthened the paper.

---

## [Referee Report · Reviewer #2 (Public review)]

Summary:

Shrews go through winter by shrinking their brain and most organs, then regrow them in the spring. The gene expression changes underlying this unusual brain size plasticity were unknown. Here, the authors looked for potential adaptations underlying this trait by looking at differential expression in the hypothalamus. They found enrichments for DE in genes related to the blood brain barrier and calcium signaling, as well as used comparative data to look at gene expression differences that are unique in shrews. This study leverages a fascinating organismal trait to understand plasticity and what might be driving it at the level of gene expression. This manuscript also lays the groundwork for further developing this interesting system.

Strengths:

One strength is that the authors used OU models to look for adaptation in gene expression. The authors also added cell culture work to bolster their findings.

Comments on revised version:

I think that the authors have made a strong revision. No other comments.

---

## [Referee Report · Reviewer #3 (Public review)]

Summary:

In their study, the authors combine seasonal and comparative transcriptomics to identify candidate genes with plastic, canalized, or lineage-specific (i.e., divergent) expression patterns associated with an unusual overwintering phenomenon (Dehnel's phenomenon - seasonal size plasticity) in the Eurasian shrew. Their focus is on the shrinkage and regrowth of the hypothalamus, a brain region that undergoes significant seasonal size changes in shrews and plays a key role in regulating metabolic homeostasis. Through comparative transcriptomic analysis, they identify genes showing derived (lineage-specific), plastic (seasonally regulated), and canalized (both lineage-specific and plastic) expression patterns. The authors hypothesize that genes involved in pathways such as the blood-brain barrier, metabolic state sensing, and ion-dependent signaling will be enriched among those with notable transcriptomic patterns. They complement their transcriptomic findings with a cell culture-based functional assessment of a candidate gene believed to reduce apoptosis.

Strengths:

The study's rationale and its integration of seasonal and comparative transcriptomics are well-articulated and represent an advancement in the field. The transcriptome, known for its dynamic and plastic nature, is also influenced by evolutionary history. The authors effectively demonstrate how multiple signals-evolutionary, constitutive, and plastic-can be extracted, quantified, and interpreted. The chosen phenotype and study system are particularly compelling, as it not only exemplifies an extreme case of Dehnel's phenotype, but the metabolic requirements of the shrew suggest that genes regulating metabolic homeostasis are under strong selection.

Weaknesses:

The results of the expression patterns are quite compelling and a number of interesting downstream hypotheses are outlined; however, the interpretation of the role of each gene and pathway identified is speculative which dampens the overall impact of the work. That said, I commend the authors on functionally testing one of the differentially expressed genes. I also commend the inclusion of that negative result.

---

## [Author Response]

The following is the authors’ response to the original reviews

**eLife Assessment**
This study presents valuable findings related to seasonal brain size plasticity in the Eurasian common shrew (Sorex araneus), which is an excellent model system for these studies. The evidence supporting the authors' claims is convincing. However, the authors should be careful when applying the term adaptive to the gene expression changes they observe; it would be challenging to demonstrate the differential fitness effects of these gene expression changes. The work will be of interest to biologists working on neuroscience, plasticity, and evolution.

We appreciate the reviewers’ suggestions and comments. For the phylogenetic ANOVA we used (EVE), which tests for a separate RNA expression optimum specific to the shrew lineage consistent with expectations for adaptive evolution of gene expression. But, as you noted, while this analysis highlights many candidate genes evolving in a manner consistent with positive selection, further functional validation is required to confirm if and how these genes contribute to Dehnel’s phenomenon. In the discussion, we now emphasize that inferred adaptive expression of these genes is putative and outline that future studies are needed to test the function of proposed adaptations. For example, cell line validations of BCL2L1 on apoptosis is a case study that tests the function of a putatively adaptive change in gene expression, and it illuminates this limitation. We also have refined our discussion to focus more on pathway-level analyses rather than on individual genes, and have addressed other issues presented, including clarity of methods and using sex as a covariate in our analyses.

**Public Reviews:**

**Reviewer #1 (Public review):**
Summary:In this paper, Thomas et al. set out to study seasonal brain gene expression changes in the Eurasian common shrew. This mammalian species is unusual in that it does not hibernate or migrate but instead stays active all winter while shrinking and then regrowing its brain and other organs. The authors previously examined gene expression changes in two brain regions and the liver. Here, they added data from the hypothalamus, a brain region involved in the regulation of metabolism and homeostasis. The specific goals were to identify genes and gene groups that change expression with the seasons and to identify genes with unusual expression compared to other mammalian species. The reason for this second goal is that genes that change with the season could be due to plastic gene regulation, where the organism simply reacts to environmental change using processes available to all mammals. Such changes are not necessarily indicative of adaptation in the shrew. However, if the same genes are also expression outliers compared to other species that do not show this overwintering strategy, it is more likely that they reflect adaptive changes that contribute to the shrew's unique traits.The authors succeeded in implementing their experimental design and identified significant genes in each of their specific goals. There was an overlap between these gene lists. The authors provide extensive discussion of the genes they found.The scope of this paper is quite narrow, as it adds gene expression data for only one additional tissue compared to the authors' previous work in a 2023 preprint. The two papers even use the same animals, which had been collected for that earlier work. As a consequence, the current paper is limited in the results it can present. This is somewhat compensated by an expansive interpretation of the results in the discussion section, but I felt that much of this was too speculative. More importantly, there are several limitations to the design, making it hard to draw stronger conclusions from the data. The main contribution of this work lies in the generated data and the formulation of hypotheses to be tested by future work.

Thank you for your interest in our manuscript and for your insights. We addressed your comments below: we now highlight the limitations of our study design in the discussion and emphasize that, while a second optimum of gene expression in shrews is consistent with adaptive evolution, we recognize that not all sources of variation in gene expression can be fully accounted for. We highlight the putative nature of these results in our revisions, especially in our new limitations section (lines 541-555).

Strengths:The unique biological model system under study is fascinating. The data were collected in a technically sound manner, and the analyses were done well. The paper is overall very clear, well-written, and easy to follow. It does a thorough job of exploring patterns and enrichments in the various gene sets that are identified.I specifically applaud the authors for doing a functional follow-up experiment on one of the differentially expressed genes (BCL2L1), even if the results did not support the hypothesis. It is important to report experiments like this and it is terrific to see it done here.

We are glad to hear that you found our manuscript fascinating and clearly written. While we hoped to see an effect of BCL2L1 on apoptosis as proposed, we agree that reporting null results is valuable when validating evolutionary inferences.

Weaknesses:While the paper successfully identifies differentially expressed seasonal genes, the real question is (as explained by the authors) whether these are evolved adaptations in the shrews or whether they reflect plastic changes that also exist in other species. This question was the motivation for the inter-species analyses in the paper, but in my view, these cannot rigorously address this question. Presumably, the data from the other species were not collected in comparable environments as those experienced by the shrews studied here. Instead, they likely (it is not specified, and might not be knowable for the public data) reflect baseline gene expression. To see why this is problematic, consider this analogy: if we were to compare gene expression in the immune system of an individual undergoing an acute infection to other, uninfected individuals, we would see many, strong expression differences. However, it would not be appropriate to claim that the infected individual has unique features - the relevant physiological changes are simply not triggered in the other individuals. The same applies here: it is hard to draw conclusions from seasonal expression data in the shrews to non-seasonal data in the other species, as shrew outlier genes might still reflect physiological changes that weren't active in the other species.There is no solution for this design flaw given the public data available to the authors except for creating matched data in the other species, which is of course not feasible. The authors should acknowledge and discuss this shortcoming in the paper.

Thank you for taking the time to provide such insightful feedback. As you noted, whiles shrews experience seasonal size changes, their environments may differ from the other species used in this experiment, leading to increased or decreased expression of certain genes and reducing our ability accurately detect selection across the phylogeny. Although we sought to control for as many sources of variation as possible, such as using only post-pubescent, wild, or non-domesticated individuals when feasible, we recognize that not all sources of variation can be fully accounted for within a practical experiment. We agree that these sources of variation can introduce both false positives and negatives into our results, and we have now highlighted this limitation within our discussion (lines 538-552).

Related to the point above: in the section "Evolutionary Divergence in Expression" it is not clear which of the shrew samples were used. Was it all of them, or only those from winter, fall, etc? One might expect different results depending on this. E.g., there could be fewer genes with inferred adaptive change when using only summer samples. The authors should specify which samples were included in these analyses, and, if all samples were used, conduct a robustness analysis to see which of their detected genes survive the exclusion of certain time points.

Thank you for this attention to detail. We used spring adults for this analysis. This decision was made as only used post pubescent individuals for all species in the analysis, and this was the only season where adult shrews were going through Dehnel’s phenomenon. We have now clarified this in both the methods and results (line 247 and line 667)

In the same section, were there also genes with lower shrew expression? None are mentioned in the text, so did the authors not test for this direction, or did they test and there were no significant hits?

We did test for decreased shrew expression compared to the rest of the species, but there were no significant genes with significant decreases. We hypothesize that there are two potential reasons for this results; (1) If a gene were to be selected for decreased expression, selection for constitutive expression of the gene across all species may be weak, and thus found in other lineages as well, or (2) decreased or no expression may relax selection on the coding regions, and thus these genes are not pulled out as we identify 1:1 orthologs. This is consistent with results provided from the original methods manuscript. Thank you for pointing out that we did not discuss this information in the text, and we now include it in our results (lines 250-251).

The Discussion is too long and detailed, given that it can ultimately only speculate about what the various expression changes might mean. Many of the specific points made (e.g. about the blood-brain-barrier being more permissive to sensing metabolic state, about cross-organ communication, the paragraphs on single, specific genes) are a stretch based on the available data. Illustrating this point, the one follow-up experiment the authors did (on BCL2L1) did not give the expected result. I really applaud the authors for having done this experiment, which goes beyond typical studies in this space. At the same time, its result highlights the dangers of reading too much into differential expression analyses.

We agree with your point, while our extensive discussion is useful for testing future hypotheses, ultimately some of the discussion may be too speculative for our readers. To amend this, we have reduced some portions of our discussion and focused more on pathways than individual genes, including removing mechanisms related to *HRH2*, *FAM57B*, *GPR3*, and GABAergic neurons. We hope that this highlights to the reader the speculative nature of many of our results.

There is no test of whether the five genes observed in both analyses (seasonal change and inter-species) exceed the number expected by chance. When two gene sets are drawn at random, some overlap is expected randomly. The expected overlap can be computed by repeated draws of pairs of random sets of the same size as seen in real data and by noting the overlap between the random pairs. If this random distribution often includes sets of five genes, this weakens the conclusions that can be drawn from the genes observed in the real data.

Thank you for highlighting this approach, it is greatly needed. After running this test, we found that observed overlapping genes were more than the expected overlap, yet not significant. We now show this in our methods (lines 277-278) and results (lines 719-720).

**Reviewer #2 (Public review):**
Summary:Shrews go through winter by shrinking their brain and most organs, then regrow them in the spring. The gene expression changes underlying this unusual brain size plasticity were unknown. Here, the authors looked for potential adaptations underlying this trait by looking at differential expression in the hypothalamus. They found enrichments for DE in genes related to the blood-brain barrier and calcium signaling, as well as used comparative data to look at gene expression differences that are unique in shrews. This study leverages a fascinating organismal trait to understand plasticity and what might be driving it at the level of gene expression. This manuscript also lays the groundwork for further developing this interesting system.

We are glad you found our manuscript interesting and thank and thank you for your feedback. We hope that we have addressed all of your concerns as described below.

Strengths:One strength is that the authors used OU models to look for adaptation in gene expression. The authors also added cell culture work to bolster their findings.Weaknesses:I think that there should be a bit more of an introduction to Dehnel's phenomenon, given how much it is used throughout.

Thank you for this insight. With a lengthy introduction and discussion, we agree that the importance of Dehnel’s phenomenon may have been overshadowed. We have shortened both sections and emphasized the background on Dehnel’s phenomenon in the first two paragraphs of the introduction, allowing this extraordinary seasonal size plasticity to stand out.

**Reviewer #3 (Public review):**
Summary:In their study, the authors combine developmental and comparative transcriptomics to identify candidate genes with plastic, canalized, or lineage-specific (i.e., divergent) expression patterns associated with an unusual overwintering phenomenon (Dehnel's phenomenon - seasonal size plasticity) in the Eurasian shrew. Their focus is on the shrinkage and regrowth of the hypothalamus, a brain region that undergoes significant seasonal size changes in shrews and plays a key role in regulating metabolic homeostasis. Through combined transcriptomic analysis, they identify genes showing derived (lineage-specific), plastic (seasonally regulated), and canalized (both lineage-specific and plastic) expression patterns. The authors hypothesize that genes involved in pathways such as the blood-brain barrier, metabolic state sensing, and ion-dependent signaling will be enriched among those with notable transcriptomic patterns. They complement their transcriptomic findings with a cell culture-based functional assessment of a candidate gene believed to reduce apoptosis.Strengths:The study's rationale and its integration of developmental and comparative transcriptomics are well-articulated and represent an advancement in the field. The transcriptome, known for its dynamic and plastic nature, is also influenced by evolutionary history. The authors effectively demonstrate how multiple signals-evolutionary, constitutive, and plastic-can be extracted, quantified, and interpreted. The chosen phenotype and study system are particularly compelling, as it not only exemplifies an extreme case of Dehnel's phenotype, but the metabolic requirements of the shrew suggest that genes regulating metabolic homeostasis are under strong selection.Weaknesses:(1) In a number of places (described in detail below), the motivation for the experimental, analytical, or visualization approach is unclear and may obscure or prevent discoveries.

Thank you for finding our research and manuscript compelling, as well as the valuable feedback that will drastically improve our manuscript. We hope that we have alleviated your concerns below by following your instructions below.

(2) Temporal Expression - Figure 1 and Supplemental Figure 2 and associated text:- It is unclear whether quantitative criteria were used to distinguish "developmental shift" clusters from "season shift" clusters. A visual inspection of Supplemental Figure 2 suggests that some clusters (e.g., clusters 2, 8, and to a lesser extent 12) show seasonal variation, not just developmental differences between stages 1 and 2. While clustering helps to visualize expression patterns, it may not be the most appropriate filter in this case, particularly since all "season shift" clusters are later combined in KEGG pathway and GO analyses (Figure 1B).- The authors do not indicate whether they perform cluster-specific GO or KEGG pathway enrichment analyses. The current analysis picks up relevant pathways for hypothalamic control of homeostasis, which is a useful validation, but this approach might not fully address the study's key hypotheses.

Thank you for this valuable feedback. We did not want to include clusters we deemed to be related to development, as this should not be attributed to changes associated with Dehnel’s phenomenon. We did this through qualitative, visual inspection, which we realize can differ between parties (i.e., clusters 2, 8, and 12 appeared to be seasonal). Qualitatively, we were looking for extreme divergence between Stage 1 and Stage 5 individuals, as expression was related to season and not development, then the average of these stages within cluster should be relatively similar. We have now quantified this as large differences in z-score (abs(summer juvenile-summer adult)>1.25) without meaningful interseason variations determined by a second local maximum (abs(autumn-winter)<0.5 and abs(winter-summer)<0.5), and added it both our methods (lines 699-702) and results (line 192).

Regarding the combination of clusters for pathway enrichment compared to individual pathways, we agree that combining clusters may be more informative for overall homeostasis, compared to individual clusters which may inform us on processes directly related to Dehnel’s phenomenon. Initially, we were tentative to conduct this analysis, as clusters contain small gene sets, reducing the ability to detect pathway enrichments. We have now included this analysis, which is reported in our methods (lines 703-704), results (lines 203-204)., and new supplemental table.

(3) Differential expression between shrinkage (stage 2) and regrowth (stage 4) and cell culture targets- The rationale for selecting BCL2L1 for cell culture experiments should be clarified. While it is part of the apoptosis pathway, several other apoptosis-related genes were identified in the differential gene expression (DGE) analysis, some showing stronger differential expression or shrew-specific branch shifts. Why was BCL2L1 prioritized over these other candidates?

We agree that our rationale for validating BCL2L1 function in neural cell lines was not clearly explained in the manuscript. We selected BCL2L1 because it is the furthest downstream gene in the apoptotic pathway, thus making it the most directly involved gene in programmed cell death, whereas upstream genes could influence additional genes or alternative processes. We have clarified this choice in the revised methods section (lines 748-750).

- The authors mention maintaining (or at least attempting to maintain) a 1:1 sex ratio for the comparative analysis, but it is unclear if this was also done for the S. araneus analysis. If not, why? If so, was sex included as a covariate (e.g., a random effect) in the differential expression analysis? Sex-specific expression elevates with group variation and could impact the discovery of differentially expressed genes.

Regarding the use of sex as a covariate, we acknowledge the concerns raised. In our evolutionary analyses, we maintained a balanced sex ratio within species when possible. EVE models handle the effect of sex on gene expression as intraspecific variation. In shrews, however, we used males exclusively, as females were only found among juvenile individuals. Including those juvenile females would have introduced age effects, with perhaps a larger effect on our results. For the seasonal data, we have now included sex as a covariate in differential expression analyses. However, our design is imbalanced in relation to sex, which we have now discussed in our methods (lines 713-714) and discussion limitations (lines 544-548).

(4) Discussion: The term "adaptive" is used frequently and liberally throughout the discussion. The interpretation of seasonal changes in gene expression as indicators of adaptive evolution should be done cautiously as such changes do not necessarily imply causal or adaptive associations.

Thank you for this insight. We have reviewed our discussion and clarified that adaptations are putative (i.e. lines 146, 285, and 332), and highlighted this in our limitations section.

**Recommendations for the authors:**

**Reviewer #1 (Recommendations for the authors):**
(1) I would recommend always spelling out "Dehnel's phenomenon" or even replacing this term (after crediting the DP term) with the more informative "seasonal size plasticity". Every time I saw "DP", I had to remind myself what this referred to. If the authors choose not to do so, please use the acronym consistently (e.g. line 186 has it spelled out).

We have replaced the acronym DP with either the full term or the more informative “seasonal size plasticity” throughout the text.

(2) Line 202: "DEG" has not been defined. Simply add to the line before.

Thank you for this attention to detail. We have added this to the line above (210).

(3) Please add a reference for the "AnAge" tool that was used to determine if samples were pubescent.

Thank you for identifying this oversight. We have now cited the proper paper in line 634.

(4) In the BCL2L1 section in the results, add a callout to Figure 2D.

We have now added a callout to Figure 2D within the results (line 234).

**Reviewer #2 (Recommendations for the authors):**
(1) Line 122: is associated? These adaptations?

Thank you for identifying that we were missing the words “associated with” here. We have fixed this in the revision.

(2) The first paragraph of the Results should be moved to the methods, except maybe the number of orthologs.

Thank you for this insight. We have removed this portion from the results section.

(3) Why a Bonferroni correction on line 188? That seems too strict.

We agree the Bonferroni correction is strict. Results when using other less strict methods for controlling false discovery rate are also not significant after correction. These corrections can be found within the data, however, we only report on the Bonferroni correction.

(4) Line 427: "is a novel candidate gene for several neurological disorders" needs some references. I see them a couple of sentences later, but that's quite a sentence with no references at the end.

We have added the proper citations for this sentence (line 524).

**Reviewer #3 (Recommendations for the authors):**
(1) Temporal Expression - Figure 1 and Supplemental Figure 2 and associated text Line176-193:- The authors report the total number of genes meeting inclusion criteria (>0.5-fold change between any two stages and 2 samples >10 normalized reads), but it would be more informative to also provide the number of genes within each temporal cluster. This would offer a clearer understanding of how gene expression patterns are distributed over time.

Unfortunately, this information is difficult to depict on our figure and would use too much space in the text. We have thus added a description of the range of genes in a new supplemental table depicting this information.

- It is unclear whether quantitative criteria were used to distinguish "developmental shift" clusters from "season shift" clusters. A visual inspection of Supplemental Figure 2 suggests that some clusters (e.g., clusters 2, 8, and to a lesser extent 12) show seasonal variation, not just developmental differences between stages 1 and 2. While clustering helps to visualize expression patterns, it may not be the most appropriate filter in this case, particularly since all "season shift" clusters are later combined in KEGG pathway and GO analyses (Fig. 1B). Using a differential gene expression criterion might be more suitable. For example, do excluded genes show significant log-fold differences between late-stage comparisons?

As previously mentioned, we have now quantified seasonal shifts as large differences in z-score (abs(summer juveniles-summer adults)>1.25) without meaningful interseason variations determined by a second local maximum (abs(autumn-winter)<0.5 and abs(winter-summer)<0.5), and added it to our methods (lines 699-702). We then follow this up with differential expression analyses as described in Figure 2.

- Did the authors perform cluster-specific GO or KEGG pathway enrichment analyses instead of focusing on the combined set of genes across the season shift clusters? While I understand that the small number of genes in each cluster may be limiting, if pathways emerge from cluster-specific analysis, they could provide more detailed insights into the functional significance of these temporal expression patterns. The current analysis picks up relevant pathways for hypothalamic control of homeostasis, which is a useful validation, but this approach might not fully address the study's key hypotheses. Additionally, no corrections for multiple hypothesis testing were applied, as noted in the results. A more refined gene set (e.g., using differential expression criteria, described above) could be more appropriate for these analyses.

We have now included cluster-specific KEGG enrichments as previously described.

(2) Differential expression between shrinkage (stage 2) and regrowth (stage 4) and cell culture targets - Figure 2 and lines195-227:- The rationale for selecting BCL2L1 for cell culture experiments should be clarified. While it is part of the apoptosis pathway, several other apoptosis-related genes were identified in the differential gene expression (DGE) analysis, some showing stronger differential expression or shrew-specific branch shifts. Why was BCL2L1 prioritized over these other candidates?

We have now included the reasoning for further validation of BCL2L1 as described above.

- The relevance of the "higher degree" differentially expressed genes needs more explanation. Although this group of genes is highlighted in the results, they are not featured in any subsequent analyses, leaving their importance unclear.

Thank you for this insight. We have removed this from the methods as it is not relevant to subsequent analyses or conclusions.

- The authors mention maintaining (or at least attempting to maintain) a 1:1 sex ratio for the comparative analysis (Line 525), but it is unclear if this was also done for the S. araneus analysis. If so, was sex included as a covariate (e.g., a random effect) in the differential expression analysis?

We have now incorporated information on sex as described above.

(3) Discussion:The term "adaptive" is used frequently and liberally throughout the discussion, but the authors should be cautious in interpreting seasonal changes in gene expression as indicators of adaptive evolution. Such changes do not necessarily imply causal or adaptive associations, and this distinction should be clearly stated when discussing the results.

Thank you for this feedback and we agree with your conclusion, while a second expression optimum in the shrew lineage is indicative of adaptive expression, we cannot fully determine whether these are caused by genetic or environmental factors, despite careful attention to experimental design. We have highlighted this as a limitation in the discussion.

(4) Minor Editorial Comment:Line 105: "... maintenance of an energy budgets..." delete "an"

We have removed this grammatical error.